

# Pashto offensive language detection: a benchmark dataset and monolingual Pashto BERT

Ijazul Haq, Weidong Qiu, Jie Guo and Peng Tang

School of Cyber Science and Engineering, Shanghai Jiao Tong University, Shanghai, Minhang, China

## ABSTRACT

Social media platforms have become inundated with offensive language. This issue must be addressed for the growth of online social networks (OSNs) and a healthy online environment. While significant research has been devoted to identifying toxic content in major languages like English, this remains an open area of research in the low-resource Pashto language. This study aims to develop an AI model for the automatic detection of offensive textual content in Pashto. To achieve this goal, we have developed a benchmark dataset called the Pashto Offensive Language Dataset (POLD), which comprises tweets collected from Twitter and manually classified into two categories: "offensive" and "not offensive". To discriminate these two categories, we investigated the classic deep learning classifiers based on neural networks, including CNNs and RNNs, using static word embeddings: Word2Vec, fastText, and GloVe as features. Furthermore, we examined two transfer learning approaches. In the first approach, we fine-tuned the pre-trained multilingual language model, XLM-R, using the POLD dataset, whereas, in the second approach, we trained a monolingual BERT model for Pashto from scratch using a custom-developed text corpus. Pashto BERT was then fine-tuned similarly to XLM-R. The performance of all the deep learning and transformer learning models was evaluated using the POLD dataset. The experimental results demonstrate that our pre-trained Pashto BERT model outperforms the other models, achieving an F1-score of 94.34% and an accuracy of 94.77%.

## INTRODUCTION

Social media has emerged as one of the most prominent and influential modes of communication and information sharing in the modern era. The usage of online social networks (OSNs) such as Twitter and Facebook has surged significantly over the past decade, with billions of users posting, sharing, and commenting on various topics. However, with the increased usage of these OSNs, there has been a rise in the prevalence of offensive language. Offensive language refers to any language used to hurt, demean, or insult an individual or group of individuals (*Cohen-Almagor, 2011*). Offensive posts can have a profound impact on OSN users who come across such content and can sometimes lead to severe cases that can foster real-world violence (*Sap et al., 2019*). The use of offensive

Corresponding author
Ijazul Haq, hanjie@sjtu.edu.cn

language not only causes harm to the targeted individuals but also harms the community as a whole. This issue has become a growing concern and has prompted many social media companies to take action by implementing policies and guidelines to monitor and remove such content. However, manual monitoring of such behavior by human moderators is not feasible due to the sheer volume of content posted every day. Additionally, the anonymity provided by social media makes it difficult to identify individuals who engage in abusive behavior. As a result, automated systems for detecting offensive content have become an essential area of research.

In recent years, there has been significant progress in the development of automated systems for the detection of offensive language on social media platforms. These systems use various NLP techniques to automate the offensive language detection process and supervise the content. Thus, promoting a safer and healthier online environment. However, most of the research so far is dedicated to the major languages such as English, Chinese, Arabic, *etc.* Low-resource languages such as Pashto, which have fewer resources for NLP, are lacking effective mechanisms for detecting and mitigating such content.

This study aims to address the issue of offensive language detection in Pashto. For this task, we have developed a benchmark Pashto offensive-language dataset which consists of Twitter tweets and comments, manually categorized into two classes "offensive" and "not-offensive". For classification, we investigated several classic deep learning sequence classifiers, such as convolutional neural networks (CNNs), and different variants of recurrent neural networks (RNNs), such as long-short term memory (LSTM) and gated recurrent units (GRU). The features we used along with these classifiers are the static word embeddings: Word2Vec, fastText, and GloVe. Moreover, we investigated the transfer learning approaches and fine-tuned XLM-RoBERTa (or XLM-R) (*Conneau et al., 2019*) on the POLD dataset, which is a pre-trained multilingual BERT model, trained using the RoBERTa (*Liu et al., 2019*) architecture. For language-specific applications, monolingual language models usually outperform multilingual models, but for the Pashto language, there is no such model publicly available at the time of writing this paper. Hence, we developed a text corpus of over 15 million words and trained a Pashto BERT model (Ps-BERT) from scratch. Similar to XML-R, we then fine-tuned Ps-BERT on the task-specific POLD dataset for offensive language detection. A high-level architecture of our work is illustrated in Fig. 1. And the key contributions of this study are summarized as follows:

- We developed a text corpus of over 15 million words and used it to pre-train the first monolingual BERT model and static word embeddings for the low-resource Pashto language.
- We developed a benchmark Pashto Offensive Language Dataset (POLD).
- We developed an NLP model for automatic detection of Pashto offensive language.

## OFFENSIVE LANGUAGE

The definition of offensive language can vary depending on cultural and societal norms, and what may be considered offensive in one context or community may not be offensive in another. However, many legal and social definitions share similar characteristics. Offensive

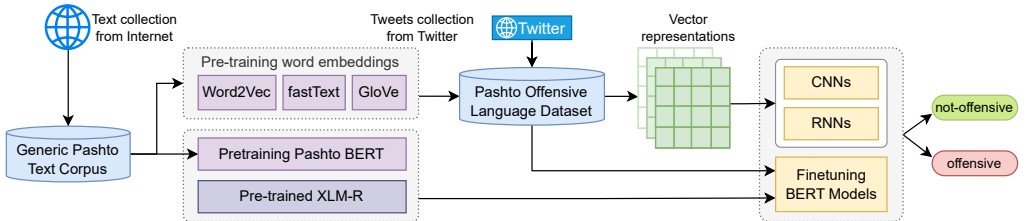

**Figure 1** A high level graphical illustration of the work.

language typically refers to any language considered to be insulting, abusive, derogatory, or discriminatory towards an individual or group based on their characteristics, such as race, ethnicity, gender, sexual orientation, religion, or disability (*Cohen-Almagor, 2011*). Offensive language includes hate speech, profanity, religious insults, and aggression, among others. Some common types of offensive language are defined as follows:

### Hate speech

According to *Allan (2013)*, hate speech is speech that attacks, insults, or threatens a particular person or group based on national origin, ethnicity, race, religion, sexual orientation, gender identity, or disability. *Davidson et al. (2017)* defined hate speech as "language used to express hate towards an individual or group intended to be derogatory, to insult or humiliate the targeted individual or members of the group". Hate speech refers to any form of speech, expression, or communication that seeks to vilify, degrade, or discriminate against someone. It demonstrates a clear intention to promote hatred and can have a profound impact on someone being targeted.

### Profanity and vulgarity

Profanity or vulgarity refers to a range of inappropriate behaviors that can manifest in various formats, such as sexually explicit content, offensive jokes, and crude sexual references (*Jay & Janschewitz, 2008*). While some individuals may view vulgar content as harmless, it can be distressing to others, particularly when used in an insulting or demeaning manner (*Mubarak, Darwish & Magdy, 2017*). The impact of vulgarity on online social networks can create an environment hostile and unwelcoming to certain individuals or groups.

### Aggression and cyberbullying

Cyberbullying can be defined as "targeted insults or threats against an individual or group" (*Zampieri et al., 2019*). It can take various forms, such as spreading rumors, making derogatory comments, blackmailing, threatening, insulting, *etc.* Cyberbullying can have more severe effects than verbal or physical bullying due to the nature of online content, which can spread quickly and viewed by a wider audience (*Dadvar et al., 2013*). Sexual harassment, which involves the use of sexual comments or gestures to harass, intimidate or offend a victim, is also a form of aggression (*Davidson et al., 2017*).

# LITERATURE REVIEW

The detection of offensive language has been a persistent problem, and significant research efforts have been directed toward this area. English has received the most attention due to the availability of language resources (*Khan et al., 2022*; *Min et al., 2023*). However, there has been a recent exploration of other languages as well, including Chinese (*Deng et al., 2022*), Arabic (*Mubarak, Darwish & Magdy, 2017*; *Alakrot, Murray & Nikolov, 2018*; *Alsafari, Sadaoui & Mouhoub, 2020*; *Althobaiti, 2022*), Hindi (*Kumar et al., 2018*), Spanish (*Aragón et al., 2019*), German (*Risch et al., 2021*) and Greek (*Pitenis, Zampieri & Ranasinghe, 2020*) to name a few. Numerous NLP techniques have been explored for this task, including machine learning (ML), deep learning (DL), and transfer learning approaches.

ML and DL techniques used for identifying offensive language are based mostly on supervised learning approaches, which involve training AI models on large datasets of labeled data. Some of the commonly used ML models for classification are Naïve Bayes (NB), support vector machines (SVMs), random forests (RF), and linear regression (LR). The DL models commonly used are CNNs and various types of RNNs such as LSTM, GRU, bidirectional LSTM (BiLSTM), and bidirectional GRU (Bi-GRU).

One of the earlier studies on toxicity and offensive language detection on social media is *Chen et al. (2012)*. They exploited the lexical, syntactic features, writing style, and structure of the cyberbullying content to identify offensive content and predict the user's intention to write offensive content. The classifiers they used are NB and SVM and reported a precision of 98.24% and recall of 94.34% on toxicity detection. *Anand et al. (2022)* used an ensemble architecture of NB, SVM, and BiLSTM for identification of objectionable content on social media. The feature selection is based on fuzzy rules and extracted using CNNs. The experiment is performed on a multilingual labeled dataset prepared from YouTube, Twitter, and Facebook. Their best-reported score is F1 of 92.5%. *Machová, Mach & Adamišín (2022)* focuses on detecting toxicity in online discussion forums using a hybrid approach that combines ML and lexicon-based methods to classify text in the Slovak language into multiple degrees of toxicity. The dataset is created from social networks in the Slovak language, and the algorithms used for classification are SVM and RF. SVM performed better and achieved an F1-score of 79%. *Lepe-Faúndez et al. (2021)* used RF, SVM, and NB classifiers with TF-IDF (term frequency-inverse document frequency) features and word embeddings for aggressiveness and cyberbullying detection in Spanish Tweets. SVM outperformed the other models and obtained an F1-score of 89.0%. To address the issue of hateful comments in the Italian language, a study presented in *Del Vigna et al. (2017)* leveraged morpho-syntactical features, sentiment polarity, and word embeddings to implement two classifiers, SVM and LSTM; and tested them on the manually annotated Italian Hate Speech Corpus. The highest accuracy reported is 85% using the SVM classifier. *Raj et al. (2021)* tested several ML models with TF-IDF features and Bi-GRU and BiLSTM models with GloVe word embeddings for cyberbullying detection. Their best score reported is F1 of 98% using GloVe with Bi-GRU. To identify offensive text in the low-resource Urdu language, which has a morphology similar to Pashto, research is

presented in *Husain & Uzuner (2022)*. Using various types of features such as BoW and Word2Vec, different ensembles of the baseline ML algorithms are used. By using alone, the RF reportedly performed best with Word2Vec features, resulting in a ROC-AUC of 87.2%.

In recent years, researchers have explored transfer learning approaches for offensive language detection. One earlier study is *Ranasinghe & Zampieri (2021)*, which utilized English datasets and applied cross-lingual transfer learning and contextual word embeddings to make predictions in low-resource languages. They employed several multilingual language models, including XLM-R and BERTm, and evaluated the systems on various benchmark datasets, including Spanish (*Basile et al., 2019*) and Hindi (*Mandl et al., 2019*); and reported that XLM-R outperforms all other methods. *Husain & Uzuner (2022)* utilized a transfer learning approach for Arabic offensive language detection in various dialects. They utilized the pre-trained BERT model for Arabic (AraBERT) and reported the best F1-score of 86% on Levantine tweets. Similarly, *Althobaiti (2022)*; *El-Alami, El Alaoui & Nahnahi (2022)* are two more examples of using transfer learning for Arabic offensive language detection. *Hussain, Malik & Masood (2022)* Proposed a binary classification model for offensive content detection in Urdu. They developed an annotated benchmark of 7,500 instances, consisting of Urdu posts from Facebook pages. Four feature extraction methods were employed, including word n-gram, bag-of-words, TF-IDF, and word2vec. The word2vec method performed the best, achieving 88.27% accuracy as a standalone model, whereas the reported accuracy for the ensemble method is 90%. *Vasantharajan & Thayasivam (2022)* focused on classifying offensive textual content on YouTube in the Tamil language, utilizing various types of pre-trained multilingual BERT models as part of ensemble learning. They found that ULMFiT and mBERT with BiLSTM yielded comparatively better results. *Benítez-Andrades et al. (2022)* focused on racism detection in Spanish tweets and investigated several deep learning and transfer learning models for this task. They compared the performance of a monolingual Spanish BERT BETO against CNNs, LSTM, and mBERT and reported the highest precision of 85.22% by using the monolingual BERT model. *Khan et al. (2022)* introduced an ensemble architecture of DL and BERT for hate speech detection on Twitter. The model takes the tweets as input and passes them through BERT, followed by an attention-aware deep convolutional layer. The convoluted representation is then passed through attention-aware BiLSTM, and finally, the tweets are labeled as normal or hateful through the SoftMax layer. *Subramanian et al. (2022)* explored various ML approaches, including SVM, LR, KNN, and transfer learning for offensive comments detection on YouTube in the low-resource Tamil language. They analyzed the three most common multilingual models, mBERT, XLM-RoBERTa, and MuRIL. Among the ML models, KNN yielded the highest accuracy of 81.65%; and XLM-R yielded the highest accuracy of 88.53% among all the ML and transformer-based models. *Mazari, Boudoukhani & Djeffal (2023)* developed an ensemble learning approach for multi-aspect hate speech detection. The pre-trained BERT Base model is combined with deep learning techniques built by stacking BiLSTMs and Bi-GRUs, with GloVe and fastText word embeddings. The experiment was performed on Jigsaw and Kaggle datasets, where the best classification result reported is a ROC-AUC of 98.63%. *Wadud et al. (2022)* developed a system for the detection of offensive language and harassment in monolingual

and multilingual texts. They combined deep CNNs and BERT. A classification layer is added on top of the encoder output, the output sequence is multiplied with the embedding matrix, and finally, the SoftMax function is used to determine the likelihood of each vector. To deal with multilingualism, they have employed collaborative multilingual and translation-based approaches. The model is evaluated on Bengali and English datasets and secured an accuracy of 91.83%. *Ali et al. (2022)* have experimented with pre-trained multilingual XLM-R and Distil-BERT for multi-class classification of hate and offensive speech in Urdu. These models are also evaluated against the baseline, classic ML models. Their best-reported results are F1-scores of 68%, 68%, and 69% for BERT, Distil-BERT, and XLM-R, respectively.

All the research work discussed above is for languages other than Pashto. To the best of our knowledge, there is currently no such study available on toxicity detection in the Pashto language. The study of *Iqbal et al. (2022)* is somewhat related to this topic, but that is primarily focused on sentiment analysis rather than offensive language detection.

## DATA ACQUISITION AND DATASET DEVELOPMENT

Data is a critical component for building effective language models. However, Pashto is a low-resource language, and electronic textual content is not abundant, making data collection a significant challenge. This study includes pre-training a BERT model and static word embeddings from scratch, which needed a substantial text corpus. Apart from the raw text corpus, a labeled dataset is also necessary to train the deep learning classifiers and fine-tune the pre-trained BERT models. In this section, we discuss the development of both prerequisites, the text corpus, and the labeled dataset.

### Pashto text corpus

To develop an efficient language model, a large corpus of text is crucial, which should be diverse and representative of the language being modeled. The original BERT was trained using a massive amount of text derived from a variety of sources, including books, websites, and Wikipedia articles. The corpus used in this study is compiled from four primary sources: news websites, Wikipedia articles, books, and Twitter posts and comments. News articles constituted the largest portion of the corpus, making up around 40% of the total. And the books included in the corpus spanned a wide range of categories, including poetry, religion, politics, fairy tales, novels, health, and academic dissertations.

The corpus has undergone several steps of pre-processing and cleaning. All the text was converted into sentences using three different sentence delimiters: the English full stop ".", the Pashto full stop "-", and the question mark. Sentences that contained words from other languages were removed, and excessively long or short sentences were also discarded. Pashto is not a standardized language, and there are no universal rules for the proper use of whitespace in the writing system. The inconsistent use of whitespace introduces noise into the text, and therefore an arbitrary Pashto text is noisier than English or other major languages. We utilized some of the techniques proposed by *Haq et al. (2023a)* to minimize the noise. The final size of our corpus is approximately 15 million tokens.

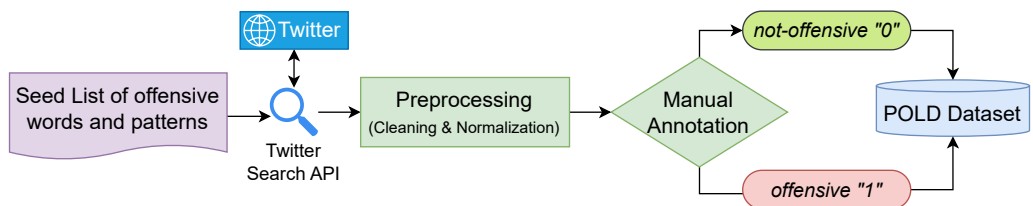

**Figure 2** **Development procedure of the pashto offensive language dataset (POLD).**

## POLD dataset

A labeled dataset is essential for supervised learning, in which the model is trained using input–output pairs. POLD is a benchmark dataset, which is developed for this study. It is a collection of tweets gathered from Twitter and manually categorized into two classes: "offensive" and "not-offensive". Figure 2 illustrates the creation procedure of the POLD dataset, and the explanation is as follows.

### Tweets collection

The first step in creating the POLD dataset was to collect raw tweets and comments written in Pashto from Twitter. However, there was a challenge; offensive tweets only make up a small portion of overall tweets, reported around 1 to 2% (*Mubarak, Darwish & Magdy, 2017*). Thus, manually annotating random tweets was not efficient. To overcome this issue, we used a seed list of offensive words to filter tweets. In order to maintain diversity in our dataset, we made the seed list large and inclusive. Another goal was to encompass all the major dialects of Pashto. And to achieve that, we made the seed list universal and included words and phrases from all the major dialects. After analyzing many tweets, we observed common patterns in offensive tweets, such as directing the speech to a specific group or person and the use of personal pronouns. In addition to offensive words, we also included these patterns in the seed list to minimize bias. The final seed list contained a total of 2.3K words and phrases, which we used to search for tweets using the Twitter Search API. The ratio of offensive tweets reached around 28% by using the words of the seed list as search keywords. We initially collected nearly 300k raw tweets between January 10 and February 10, 2023. These tweets were then pre-processed to compile the dataset.

### Pre-processing

During the pre-processing stage, the tweet corpus underwent several cleaning and normalization steps, which involved removing HTML tags, usernames, URLs, and other special characters. While diacritics are not part of the Pashto writing system and do not contribute to the meaning, they exist in informal textual content on social media. Therefore, we removed all the diacritics and normalized the letters to their purest form. Additionally, words from other languages were removed, and digits were normalized to the Pashto format. Duplicate tweets were removed, and tweets with fewer than 10 or more than 150 characters were discarded. After pre-processing, the size of the corpus was reduced to 70K tweets. POLD is a manually labeled dataset, and to maintain quality, we reduced its size

to minimize human effort in manual annotation. We randomly selected 35K (50%) of the tweets for manual annotation, and the rest were discarded. Some noisy tweets that went undetected during the pre-processing stage were identified and eliminated during manual annotation.

### Manual annotation

Involving human annotators in creating benchmark, labeled datasets is inevitable. For manual annotation, it is commonly preferred to use an online crowdsourcing platform. However, for the Pashto language, we were unable to find skilled and qualified annotators online. Therefore, we hired four professionals for the manual annotation. All the participants are native Pashto speakers and university graduates who were paid for this task to speed up the time-consuming annotation process without compromising the quality. Hence, the manual annotation was carried out by a total of five participants, including one of the authors, who also is a native Pashto speaker. To ensure that the resulting models are accurate and robust, we aimed to build a diverse dataset free from bias toward any specific individual, group, or ideology. Therefore, we ensured that each participant adhered to the widely agreed-upon definitions of offensive language discussed in Section 'Offensive Language', and the guidelines mentioned in OffensEval2019 (*Zampieri et al., 2019*). Tweets containing any type of offensive language, such as hate speech, cyberbullying, aggression, abuse, or profanity, were assigned the label "1" (offensive), and the rest (neutral or positive tweets) were assigned the label "0" (non-offensive). The manual annotation task was accomplished in two stages.

In the first stage of annotation, three participants (one author and two paid professionals) participated in the annotation process. The complete corpus was individually tagged by each annotator, without knowing the decision of other annotators. This way, each tweet was labeled three times, once by each annotator. After the annotation was completed, the votes were tallied. In the first stage of annotation, 31,300 tweets (around 91%) were annotated with 100% inter-annotator agreement. These tweets exhibit a clear polarity and can easily be differentiated without deeper scrutiny. We assumed that no matter how many annotators labeled these tweets, the inter-annotator agreement would be (or come close to) 100%. Therefore, these tweets were categorized based on the decision of these three annotators and included as the first entries in the POLD dataset.

In the first stage of annotation, 9% of the tweets did not achieve 100% inter-annotator agreement. In the second stage of annotation, these tweets were labeled by two additional annotators (paid participants who did not take part in the first round). Thus, somewhat ambiguous tweets were annotated (voted) by a total of five participants. The final decision regarding the status of these tweets was made by a majority vote. Figure 3 is a word frequency diagram, which shows the most frequent words in the POLD dataset.

### Dataset summary

A summary of the POLD dataset and the splitting method is presented in Table 1. The final size of the dataset is 34,400, with 12,400 instances labeled as offensive and 22,000 labeled as non-offensive. A comparison of some popular benchmark datasets for offensive language
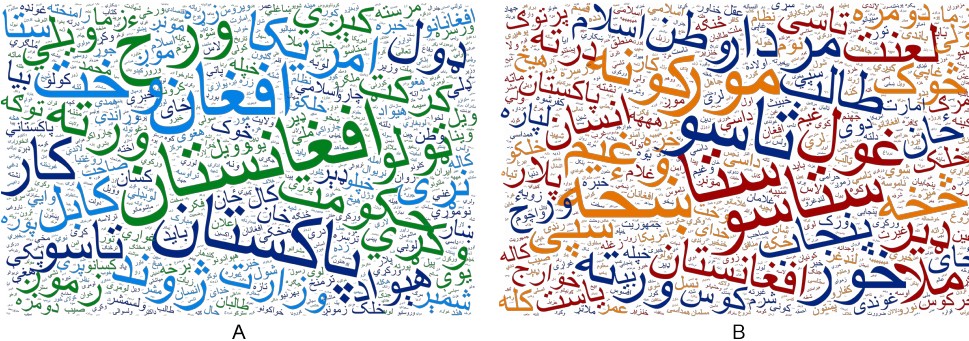

**Figure 3** Word frequency diagrams of the most frequent words in POLD dataset (A) in normal (not-offensive) tweets and (B) in offensive tweets.

**Table 1** Summary of the POLD dataset.

| Split | Offensive | Not-offensive | Total |
|---|---|---|---|
| Train | 9,920 | 17,600 | 27,520 |
| Validation | 1,240 | 2,200 | 3,440 |
| Test | 1,240 | 2,200 | 3,440 |
| Total | 12,400 | 22,000 | **34,400** |

detection is given in Table 2, where we can see that POLD has a competitive size despite being a dataset of low-resource language. The average length of tweets in the POLD dataset is 18 tokens, which is comparatively smaller than the average sentence size in the Pashto text corpus of around 26 tokens. The reason is that most tweets (especially comments) on Twitter are shorter, sometimes only one or two words. The dataset is imbalanced, where the "not-offensive" class is nearly three times larger than the "offensive" class; however, this is the case in the real world, where only a fraction of user-generated content is offensive. Nevertheless, an automatic classifier needs to be able to handle the issue of an imbalanced dataset. The dataset is split into three portions: training set (80%), validation set (10%), and test set (10%); where, in each split, we have ensured a proportional representation of both offensive and non-offensive tweets to maintain a balanced class distribution and minimize potential skewness. The same portions of the dataset: training, validation, and test are used uniformly across all the models for training, evaluation, and testing, respectively.

## METHODS

In this study, we have investigated two methods for Pashto offensive language detection, (i) deep learning methods and (ii) transfer learning methods. A graphical illustration of these methods is presented in Fig. 4, and the explanation is as follows.
**Table 2  Comparison of POLD with some other popular datasets.**

| Reference | Language | Classes | Total size |
|---|---|---|---|
| *Mubarak, Darwish & Magdy (2017)* | Arabic | Obscene: 2%<br>Offensive: 79%<br>Clean: 19% | 32,000 |
| *Deng et al. (2022)* | Chinese | Offensive: 18,041<br>Not-offensive: 19,439 | 37,480 |
| *Pitenis, Zampieri & Ranasinghe (2020)* | Greek | Obscene: 29%<br>Not-offensive: 71% | 4,779 |
| *Özberk & Çiçekli (2021)* | Turkish | Offensive: 6,845<br>Not-offensive: 28,429 | 35,284 |
| *Zampieri et al. (2019)* | English | Offensive: 4,640<br>Not-offensive: 9,460 | 14,100 |
| *Davidson et al. (2017)* | English | Hate: 1,430<br>Non-hate: 4,163 | 5,593 |
| *Pereira-Kohatsu et al. (2019)* | Spanish | Hate: 1,567<br>Non-hate: 4,433 | 6,000 |
| *Ibrohim & Budi (2019)* | Indonesian | Hate: 5,561<br>Non-hate: 7,608 | 13,169 |
| *Ataei et al. (2022)* | Persian | Offensive: -<br>Not-offensive: - | 10,563 |
| **POLD (Ours)** | Pashto | Offensive: 12,400<br>Not-offensive: 22,000 | 34,400 |

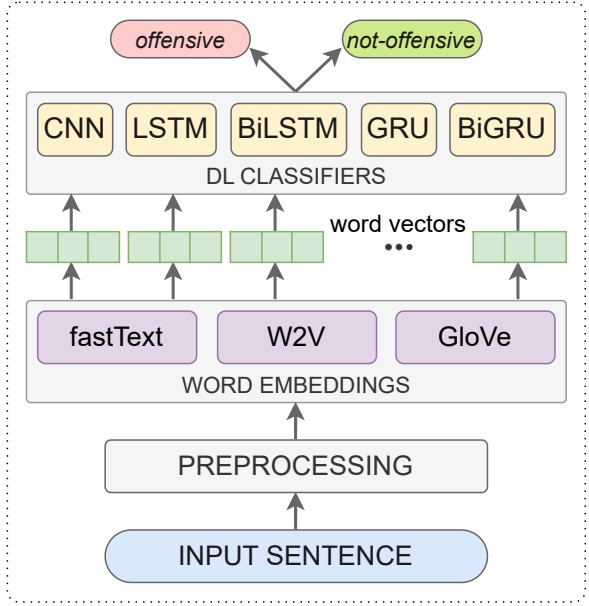

Deep Learning Methods

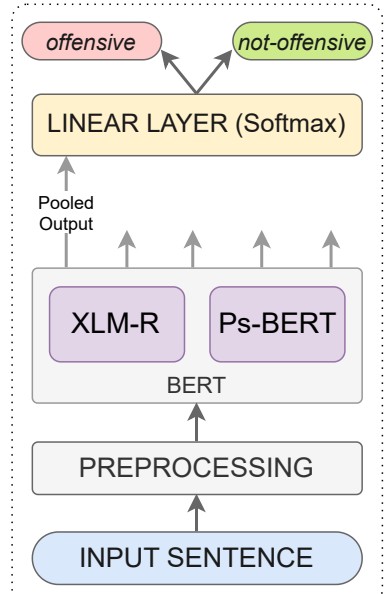

Transfer Learning Methods

**Figure 4  Graphical illustration of the deep learning and transfer learning methods for Pashto offensive language detection.**

## Deep learning methods

In the deep learning approach, we pre-trained static word embeddings and conducted experiments to examine the performance of baseline neural network-based classifiers for Pashto offensive language detection.

### Word embeddings

Word embeddings involve mapping vocabulary tokens/words to real number vectors. The objective is to learn a distributed representation of words based on their co-occurrence in a large text corpus. A neural network is trained to capture the syntactic and semantic meaning of words, which allows it to associate words with similar vector representations based on their contexts. The resulting vector representations are used as input for other ML models. NLP researchers usually utilize pre-trained word embeddings trained on extensive corpora containing millions or billions of tokens, but for Pashto, no word embedding is publicly available, except fastText. Hence, we trained the three popular word embeddings Word2Vector, fastText, and GloVe from scratch.

Word2Vec has two variants: continuous bag-of-words (CBOW) and skip-gram. CBOW predicts the target word given the context words, while skip-gram predicts the context words given the target word. CBOW is faster and works well with frequent words, while skip-gram is more efficient and produces more accurate results on medium-sized corpora, making it the preferred approach. FastText extends Word2Vec-type models with sub-words. It breaks down words into character n-grams or sub-words and learns vector representations for these sub-words. These sub-word vectors are added together to build the final word vector. By exploiting the sub-word information, fastText can handle OOV errors; because if the word is not in the vocabulary, its sub-words might be. This approach is also useful in obtaining representations for obfuscated or misspelled words. On the other hand, GloVe embeddings are trained on a global word-word co-occurrence matrix. It captures the statistical relationships between words based on their frequency of co-occurrence in the text corpus, explicitly modeling the global context of words and allowing it to capture more nuanced relationships. GloVe has been shown to outperform Word2Vec and fastText on various NLP applications, such as word similarity tasks.

For training the Pashto word embeddings, we used our Pashto text corpus as the training data. For all three models, we kept most of the hyper-parameters uniform. We fixed the size of the embedding vector to 100, the window size was 5, and the minimum count was 2, which is the minimum frequency needed for a word to be included in the final vocabulary. We chose the skip-gram architecture for Word2Vec and fastText, and each model is trained for five epochs. The GloVe model is trained using the GloVe package, while Word2Vec and fastText are trained using the Gensim and fastText Python libraries.

### Neural networks

Neural networks are commonly used for text classification to learn complex patterns and relationships in textual data. In this study, we considered five types of neural networks for the experiment, CNNs, and four types of RNNs (LSTM, BiLSTM, GRU, and Bi-GRU).

CNNs have traditionally been used for image recognition and classification, but they can also be applied to text classification tasks. In text classification, the text data is represented

as a two-dimensional matrix where the rows correspond to words, and the columns correspond to the embedding dimensions. The CNN architecture applies convolutional filters over the sequence of word embeddings to extract relevant features and then uses max-pooling to capture the most important features. CNNs can capture local dependencies and relationships between adjacent words and learn hierarchical features at different levels of abstraction.

RNNs are a special type of artificial neural network, which are designed to handle sequential data and are well-suited for text classification tasks. In RNNs, the input text is fed into a recurrent layer, which can capture the context of each word. Unlike traditional feed-forward neural networks, RNNs have a feedback loop that allows information to persist from one-time step to the next.

LSTM is a commonly used type of RNNs that addresses the vanishing gradient problem often encountered in classic RNNs. The vanishing gradient problem occurs when the gradients used to update the weights in the network become very small, making it difficult to learn long-term dependencies. LSTM employs memory cells that can store and update information over long time steps. Each memory cell is controlled by three gates: the input gate, output gate, and forget gate, which regulate the flow of information in and out of the cell. A BiLSTM architecture is comprised of two LSTMs, one of which takes the input forward and the other backward. It captures long-range dependencies from both directions of a sequence. The outputs of both LSTM layers are merged to produce the final output. By combining both left and right contexts, BiLSTM can model more complex dependencies between words in some cases.

GRU is a neural network architecture similar to LSTM but with only two gates: a reset gate and an update gate. Unlike the LSTM, the GRU does not have a separate output gate, since the hidden state serves as both the cell state and the output. Similarly, Bi-GRU is a variation of the GRU that processes the input sequence in both forward and backward directions, like the BiLSTM.

### Training the neural networks

The primary components of all neural networks are the embedding layer, hidden layer, and output layer. The Embedding layer serves as the first hidden layer and is a matrix of size $m \times n$, where m is the size of vocabulary in the embedding matrix and n is the maximum length of sequences (tweets), fixed at 64 tokens. To prevent overfitting, we used a dropout of 0.2. The output layer employs the Sigmoid activation function and the Adam optimizer and uses cross-entropy loss to predict text labels. Apart from these primary (common) components, each model has its adaptation and hyper-parameters settings, discussed as follows.

For the CNN model, we constructed a 1D convolutional layer with 100 filters and a kernel of size 4. The subsequent layer is max-pooling with default values, followed by a dropout layer, and then the output layer to assign a category to each tweet. The LSTM model comprises one LSTM layer with 100 units and a dropout layer, followed by a classification layer that predicts the category of tweets. A similar architecture is used for the GRU model also, while the LSTM layer is replaced by a GRU layer. To build the Bidirectional LSTM,

**Table 3  Implementation details and hyper-parameters of the neural networks.**

| Model | Architecture and Hyper-parameters |
|---|---|
| CNN | 1D convolutional layer ($F = 100$, $K = 3$) |
| | Max-pooling layer with default values |
| | Dropout layer (dropout $= 0.2$) |
| | Fully connected layer |
| | Output layer with Sigmoid function |
| LSTM | 1 LSTM layer (100 hidden unit) |
| | Dropout layer (dropout $= 0.2$) |
| | Fully connected layer |
| | Output layer with Sigmoid function |
| BiLSTM | 1 BiLSTM layer with 100 hidden units |
| | Dropout layer (dropout $= 0.2$) |
| | Fully connected layer |
| | Output layer with Sigmoid function |
| GRU | 1 GRU layer (100 hidden unit) |
| | Dropout layer (dropout $= 0.2$) |
| | Fully connected layer |
| | Output layer with Sigmoid function |
| BiGRU | 1 BiGRU layer with 100 hidden units |
| | Dropout layer (dropout $= 0.2$) |
| | Fully connected layer |
| | Output layer with Sigmoid function |

we construct one BiLSTM layer with 100 hidden units. The output vectors are flattened and fed to the classification layer. Similarly, the Bi-GRU is built using a Bi-GRU layer with the uniform configuration of BiLSTM. We used a batch size of 32 and trained each model for 5 epochs. The implementation details and hyper-parameters setup of the models are summarized in Table 3, and the loss curves of the training and validation are plotted in Fig. 5.

## Transfer learning methods

Conventional machine learning generally involves training a model from scratch using a large dataset, whereas transfer learning uses a pre-trained model as a starting point to solve a new task. For major languages such as English, Chinese, or Arabic, researchers have pre-trained large language models on huge corpora that can be used off-the-shelf and fine-tuned for a specific task, such as offensive language detection. However, for Pashto, to our knowledge, no such pre-trained language model is publicly available, while they are essential for NLP research nowadays. To incorporate transfer learning for Pashto offensive language detection, we employed two approaches. Firstly, we fine-tuned a pre-trained multilingual BERT model, XML-R, using the POLD dataset. But, multilingual models are generally less effective for language-specific tasks; therefore, in the second approach, we pre-trained a Pashto monolingual BERT model from scratch and then fine-tuned it the same way as XLM-R.

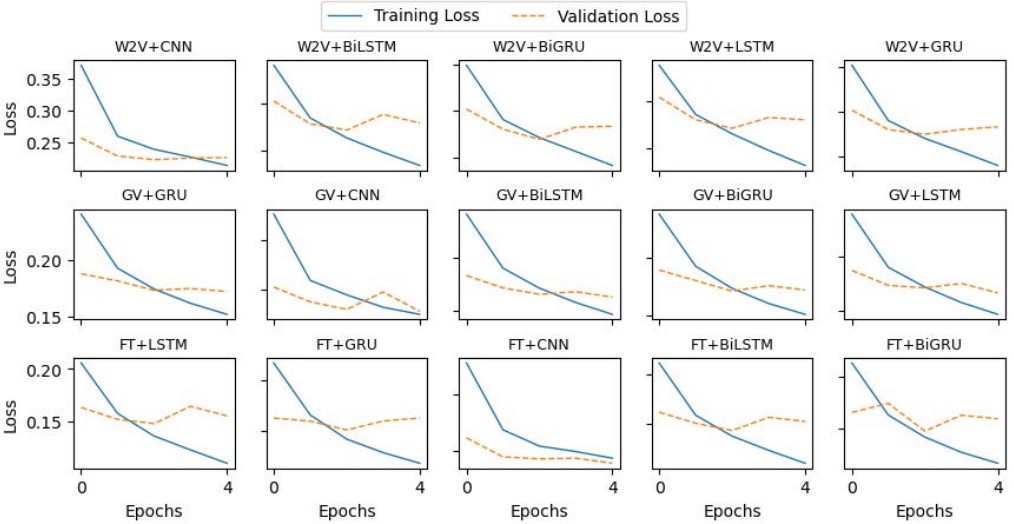

**Figure 5  Loss curves of the deep learning models during training.**

### Pre-trained multilingual models

There are several multilingual pre-trained models available, though most of them are missing the Pashto language. The pre-trained language model we investigated in this study is the XLM-R, which has been pre-trained in more than 100 languages, including Pashto. The XLM-R is built upon the RoBERTa architecture, which is a variant of the BERT model. And, like RoBERTa, XLM-R is pre-trained on a large corpus of text using a masked language modeling (MLM) task. However, unlike RoBERTa, it is trained on text from multiple languages. XLM-R is trained on 2.5TB of CommonCrawl data in 100 languages simultaneously. This model outperforms other multilingual models on many NLP tasks, demonstrating its effectiveness at learning cross-lingual representations. We fine-tuned the base version of XML-R, which has 12 layers, 768 hidden states, 12 attention heads, and 270 million parameters.

### Pashto BERT: pre-training from scratch

For pre-training the monolingual Pashto BERT (Ps-BERT) model from scratch, we utilized the BERT Base (*Devlin et al., 2018*) variant, which has 12 layers, 768 hidden states, 12 attention heads, and 110 million parameters. It can handle up to 512 tokens in an input sequence. The beginning of a text sequence is indicated by the (CLS) (classification) token, and the (SEP) (separator) token is used to indicate the end. For each token in the input sequence, the BERT model generates a corresponding vector representation in each encoder layer, and the (CLS) token representation is the sentence representation. For training, we employed the MLM task, which involves randomly masking some input tokens and training the model to predict the original token from its context. The pre-training procedure of the Ps-BERT is illustrated in Fig. 6.

**Data:** Training data is the Pashto text corpus consists of over 15 million words, discussed in 'Data Acquisition and Dataset Development'.

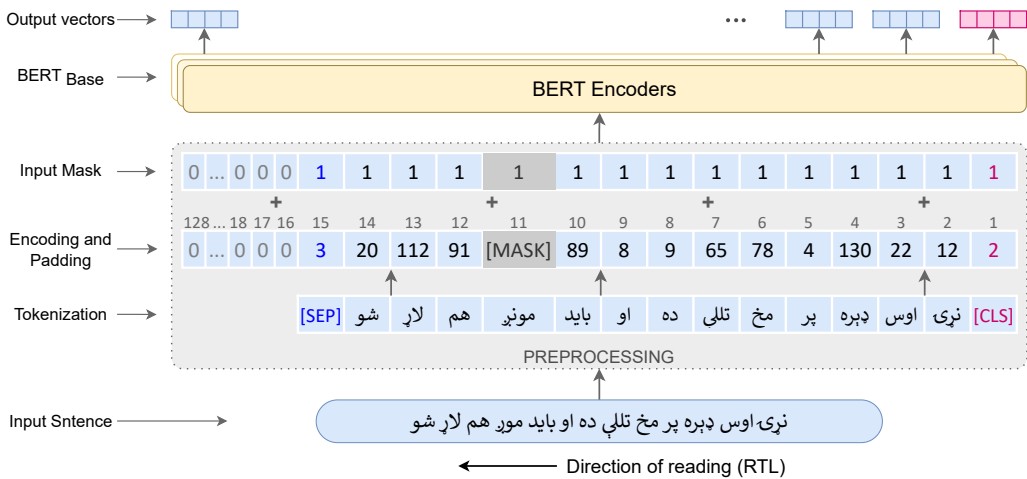

**Figure 6** **Pre-training procedure of the Pashto BERT Model.**

**Tokenization:** To tokenize the input sentences (tweets), we used the BERT WordPiece tokenizer (*Schuster & Nakajima, 2012*), which is the recommended tokenizer for the BERT Base model, and the vocabulary size was fixed at 30K words. Two special tokens, (CLS) and (SEP) were added to the beginning and end of the sequences, respectively. To enable the model to differentiate between original and padded tokens in the input sequences, we employed an attention mask to generate a vector of 1s and 0s for each input sequence, where 0s indicate the padded tokens and 1s indicate the original tokens.

**Hyper-parameters and Optimization:** We used a batch size of 32 sequences, where the maximum sequence length was fixed at 128 tokens. Sequences with more than 128 tokens were truncated, while shorter ones were post-padded. We used the Adam optimizer with a learning rate of 1e−4 and performed a linear warmup schedule with the first 10K steps. We used $\beta 1 = 0.9$ and $\beta 2 = 0.999$, L2 weight decay of 0.01 and epsilon of 1e−8.

**Implementation and Training:** We implemented the model architecture and training pipeline using PyTorch and the Huggingface transformers library. And the model was trained on a cloud GPU—NVIDIA Tesla P100 that took over 1 h to complete.

### Fine-tuning

Fine-tuning is the process of adapting the pre-trained models to a specific downstream task by fine-tuning its parameters on a labeled dataset. Fine-tuning a BERT model involves adding a classification layer on top of the pre-trained model and training the model using back-propagation and an optimizer. For text classification tasks, BERT takes the final hidden state of the (CLS) token as the representation of the whole sequence.

We fine-tuned both the pre-trained models, XLM-R and Ps-BERT, using the task-specific POLD dataset. Fine-tuning involves the same pre-processing steps as in pre-training from scratch. The tweets from the dataset were tokenized into a sequence of tokens adding special tokens (CLS) and (SEP) to mark the beginning and end of the sequences, respectively. However, the tokenizers used by RoBERTa and BERT architectures are different, where

**Table 4  Hyper-parameters setup for fine-tuning XLM-R and Ps-BERT.**

| Hyper-parameters | XLM-R | Ps-BERT |
|---|---|---|
| Learning rate | 2e−5 | 5e−5 |
| Adam $\beta1$, $\beta2$ | (0.9, 0.999) | (0.9, 0.999) |
| Adam Epsilon | 1e−8 | 1e−8 |
| Sequence length | 100 | 100 |
| Batch size | 16 | 16 |

**Figure 7  Loss curves of fine-tuning the XLM-R and Ps-BERT models.**

RoBERTa expects the input sequence to be tokenized using the SentencePiece tokenizer (*Kudo & Richardson, 2018*), while BERT utilizes the WordPiece, as mentioned earlier. XLM-R is based on the RoBERTa architecture; therefore, we used the SentencePiece tokenizer to tokenize sequences for it. For fine-tuning, most of the model hyper-parameters are the same as in pre-training, with the exception of learning rate and sequence length. Table 4 shows the optimal values of hyper-parameters for each model that we chose after an exhaustive search. The loss curves of models' training and validation are plotted in Fig. 7.

## EXPERIMENTAL RESULTS AND EVALUATION

This section presents the performance evaluation results of all the models we investigated for detecting offensive language in Pashto. We used a GPU-facilitated Kaggle platform to conduct the experiments. Data preparation and pre-processing was performed using Pandas and Numpy libraries. The deep learning models were implemented with Keras, and transformers were built using the Huggingface library and PyTorch, while scikit-learn packages were utilized for evaluation. The dataset was split into three portions, training, validation, and test set, detailed in 'Dataset Summary' and Table 1. The training set was used to train the models, and the evaluation set was used to fine-tune the hyperparameters and prevent overfitting during the training process. Finally, the test set was used to evaluate the models and test their generalization capability on unseen data.

## Evaluation matrices

The performance of machine learning models is generally evaluated using precision, recall, F-score, and accuracy. In this study, we also used these four metrics to evaluate the performance of our models and compare the results. These metrics are particularly useful when the dataset is imbalanced, like POLD. In the context of this article, precision refers to the percentage of correctly classified offensive tweets out of the total classified offensive tweets, as shown in Eq. (1). The recall represents the percentage of correctly labeled hate tweets out of the total labeled offensive tweets and its mathematical formulation defined in Eq. (2). The F1-score is a harmonic mean of precision and recall, providing a balanced evaluation of the classifier performance, which can be calculated using the formula in Eq. (3). Finally, accuracy represents the ratio of correctly classified tweets to all the classified tweets, as defined in Eq. (4).

$$Precision = \frac{True\ Positives}{(True\ Positives + False\ Positives)} \tag{1}$$

$$Recall = \frac{True\ Positives}{(True\ Positives + False\ Negatives)} \tag{2}$$

$$F1 - score = 2 \times \frac{Precision \times Recall}{Precision + Recall} \tag{3}$$

$$Accuracy = \frac{True\ Positives + True\ Negatives}{Total\ Tweets} \tag{4}$$

## Comparison of all the models

Table 5 presents a comparative performance evaluation of the various models, along with a graphical representation in Fig. 8. The experimental results demonstrate that the transformer-based models perform superior to traditional deep learning models. Among all the models we investigated, the fine-tuned Ps-BERT model yields the best performance and achieves an F1-score of 94.34% with an accuracy of 94.77%. While the XLM-R, the other transformer-based model, performs slightly lower than Ps-BERT, it still surpasses the neural networks-based models. Although the difference in performance between XLM-R and Ps-BERT is not significantly large, there is a significant difference between the resources utilized to pre-train these models. XLM-R has been pre-trained on a huge corpus of billions of tokens and consumed several GPU hours. In contrast, the Ps-BERT model was trained on a corpus of 15 million tokens, taking around one hour on a single GPU chip. It is usually the case that monolingual language models tend to outperform multilingual models on language-specific tasks.

Concerning the neural network models, the results indicate that the RNN models perform better than CNNs. It can be seen that, in all the RNNs, the LSTM classifier with fastText embeddings outperforms the other models by achieving an F1-score of

**Table 5  Comparison of all the models.**

| Model | Precision (%) | Recall (%) | Accuracy (%) | F1-score (%) |
|---|---|---|---|---|
| CNN+Word2Vec | 91.02 | 87.85 | 90.29 | 89.08 |
| CNN+fastText | 92.25 | 90.80 | 92.24 | 91.44 |
| BiGRU+Word2Vec | 92.59 | 90.67 | 92.33 | 91.50 |
| BiLSTM+Word2Vec | 93.12 | 91.29 | 92.85 | 92.09 |
| GRU+Word2Vec | 93.23 | 91.38 | 92.94 | 92.18 |
| CNN+GloVe | 93.09 | 91.56 | 92.97 | 92.24 |
| LSTM+Word2Vec | 93.41 | 91.82 | 93.23 | 92.52 |
| BiGRU+GloVe | 93.48 | 92.13 | 93.40 | 92.74 |
| GRU+GloVe | 93.65 | 92.03 | 93.43 | 92.75 |
| BiLSTM+GloVe | 93.33 | 92.27 | 93.40 | 92.76 |
| LSTM+GloVe | 93.20 | 92.41 | 93.40 | 92.78 |
| BiLSTM+fastText | 93.76 | 92.06 | 93.49 | 92.81 |
| GRU+fastText | 93.66 | 92.15 | 93.49 | 92.82 |
| BiGRU+fastText | 93.43 | 92.30 | 93.46 | 92.82 |
| LSTM+fastText | 93.88 | 92.43 | 93.72 | 93.08 |
| XLM-R | 93.98 | 94.05 | 94.48 | 94.01 |
| Ps-BERT | 94.22 | 94.47 | 94.77 | **94.34** |

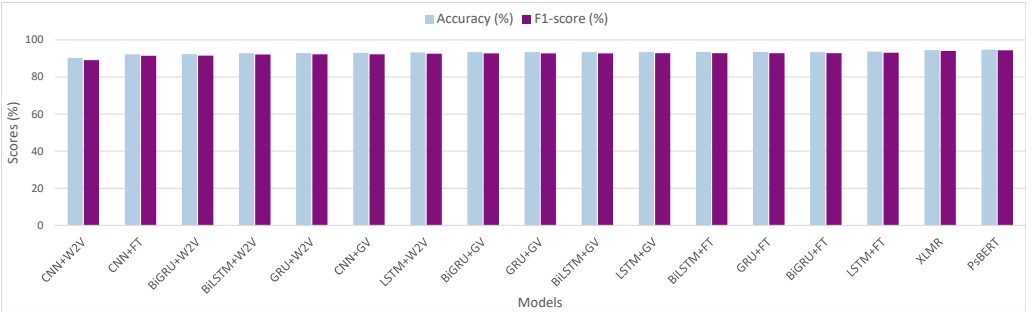

**Figure 8  Graphical representation of models' performance in terms of accuracy and F1-score.**
Word2Vec, GloVe and fastText are represented by W2V, GV and FT respectively.

93.08% with an accuracy of 93.72%. In bidirectional RNNs, BiGRU performs the best with an F1-score of 92.82% and accuracy of 93.46%, whereas the unidirectional GRU also achieved quite similar scores. On the downside, the CNNs-based model with Word2Vec embeddings exhibits the lowest performance among all the neural networks we examined, with an F1-score of 89.08% and an accuracy of 90.29%. The results also demonstrate that the difference among the RNNs is not very large, and similarly, the difference between the bidirectional and unidirectional RNNs is also not significant.

## Comparison of the static word embeddings

Figure 9 illustrates the performance comparison of different static word embeddings used with various neural network classifiers. The obtained results show that fastText outperforms

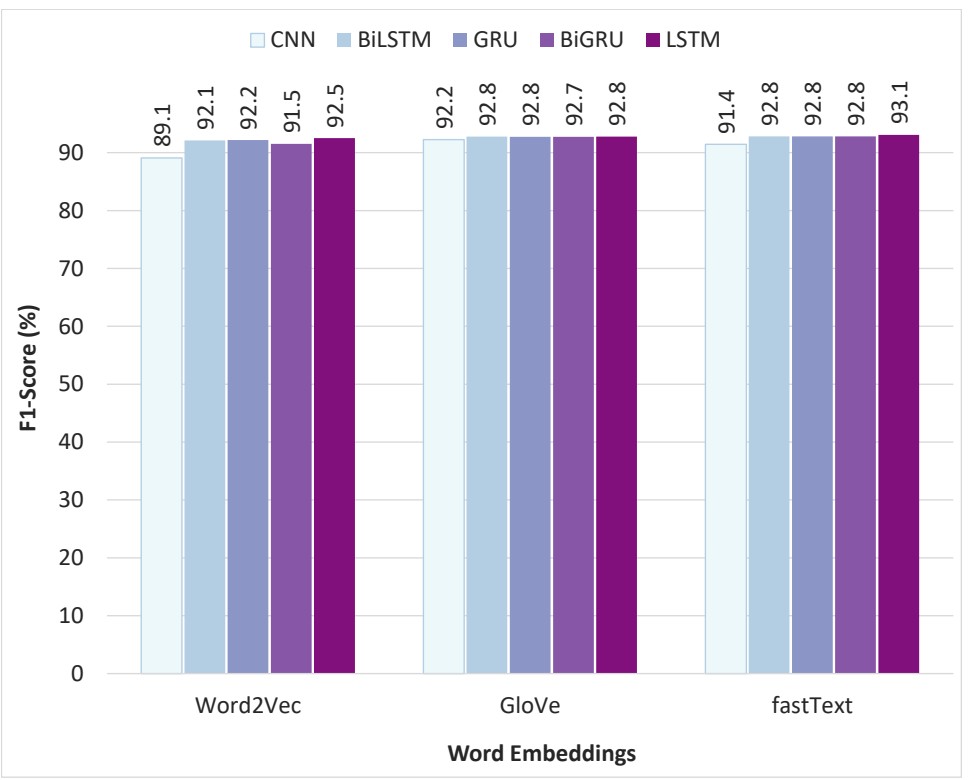

**Figure 9** Comparison of the static word embeddings.

the other word embedding models and achieves the highest F1-score of 93.08% when fed to the LSTM classifier. Additionally, fastText performs uniformly well on the other classifiers as well. The fastText model uses sub-word (or n-gram) level tokenization, which is particularly beneficial for the task of offensive language detection. The reason is that users on social media platforms often use alterations of the words or write half words instead of the full form, especially when writing inappropriate words. For example, on English social media, words like "b!tch", "c#ck", "f*ck" *etc.*, are commonly used, and the same convention exists in Pashto social media also. This way of writing often leads to the problem of OOV errors in the other two word embedding models: Word2Vec and GloVe. In contrast, fastText exploits the sub-word information, so if a word is not present in the vocabulary, its sub-words might be, which is useful in obtaining representations for altered, misspelled, or half-words. Regarding the other two word embedding techniques, Word2Vec performed poorly, while GloVe achieved comparatively satisfactory results and slightly lagged behind fastText.

## Error analysis

The results presented in Table 5 confirmed that the fine-tuned monolingual Pashto BERT is the best-performing model in identifying offensive Pashto language. However, to gain more in-depth insights, we performed a detailed analysis of the individual models' errors using confusion matrices. A confusion matrix summarizes the number of correct

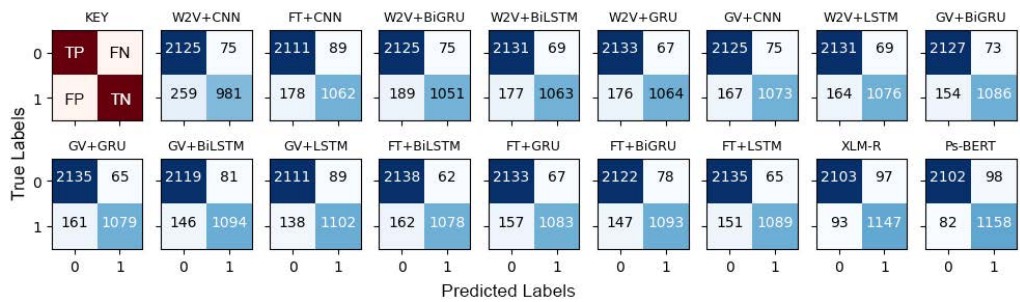

**Figure 10 Confusion Matrices of the models, where "0" represents "not-offensive" and "1" represents the "offensive" class.** The first sub-figure is the key to interpreting the confusion matrices.

and incorrect predictions, with value count and breakdown by each class. It provides insights into the model errors and the types of errors. For instance, the confusion matrix provides a clear breakdown of the model performance in terms of true positives (TP), true negatives (TN), false positives (PP), and false negatives (FN). Figure 10 shows the confusion metrics of the models, where the label "1" and label "0" represents the "offensive" and "non-offensive" classes respectively. The major diagonal of the confusion matrix shows the correct predictions of the model, and the minor diagonal shows the incorrect predictions. There are a total of 3,440 instances in our test dataset, where the worst classification model in this study is the CNN+Word2Vec, which has incorrectly classified 334 instances. In contrast, Ps-BERT is the best classifier, which has incorrectly classified only 180 instances. The ROC-AUCs (receiver operating characteristic–area under the curve) of the model are depicted in Fig. 11. This metric is widely employed in binary classification tasks, especially when dealing with imbalanced datasets. The ROC-AUC plots the true positive rate (TPR) against the false positive rate (FPR). The AUC represents the area under the curve, which is a measure of the model's capacity to differentiate between the two classes at a given threshold, 0.5 in this context. The AUC ranges from 0 to 1, with a higher value indicating better model performance, where our best-performing model, PsBERT, yielded an AUC of 94.47

In Table 6, we present a selection of the tweets misclassified by the Ps-BERT model. Upon manual inspection of these tweets, a significant portion of the false-positive instances were found to be poetic. One possible reason is that poetry often includes phrases that are directed toward someone, and the model has learned to associate such directed phrases and second-person pronouns with offensive content. Anyhow, this observation reveals a limitation of the model performance. But hopefully, this issue can be addressed by expanding the dataset size and incorporating a larger and more diverse range of poetic examples.

## CONCLUSION

In this study, we investigated the effectiveness of neural network-based models and the potential of transfer learning techniques for detecting toxic Pashto language. Specifically,

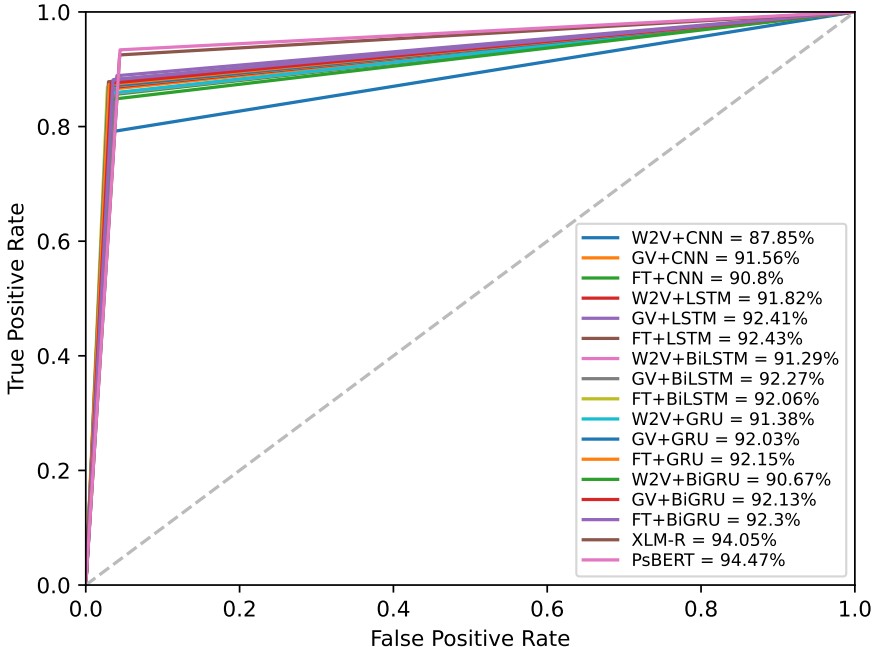

**Figure 11** ROC-AUC curves of the models at a threshold of 0.5.

**Table 6** Example sentences that are misclassified by the Ps-BERT model, where 0 and 1 represents the not-offensive and offensive classes respectively.

| Sentences | True | Predicted |
|---|---|---|
| غازيان ځي خو کارنامې يې ژوندۍ وي | 0 | 1 |
| زما دي شال په پرې زوړ کرو ستا د کويلي د کانه نه خيجي لاسونه (poetic) | 0 | 1 |
| خوارج د مسلمانانو عامه ځايونه هدف ګرځوي او بريدونه ورباندې کوي | 0 | 1 |
| کله کله می حالات داسی مجبور کړي د تپوس بچي ته وايم چی شاهين يې (poetic) | 0 | 1 |
| خپل منزل ته رسيدل به درته ګران شي که په لاره غپيدلو سپو ته غوږ شې (poetic) | 0 | 1 |
| ولا نو سخت خوند کوي چې اسلامي نظام وي او باډيګاردان درنه چاپېپپېره وي | 0 | 1 |
| سپيه مالياتی نظام حضرت عمر رض رامنخته کړ | 1 | 0 |
| پريزده مره تول به دې ۰۹ ثانتی قد وي خبري دي ګوره | 1 | 0 |
| نن درته هندوان اوږه خيرات کی ليژلی سپين سترګی شرم وکړه | 1 | 0 |
| خور دې راکوي چپ داسپ پلتنه دې پيل کړې هو همدا زه يم | 1 | 0 |
| چپ عمر يي له ۰۲ کلو پورته وي هغه نسل بايد تول ورک شي | 1 | 0 |

we tested five neural networks: CNN, LSTM, GRU, BiLSTM, and BiGRU, across three word embedding schemes: Word2Vec, fastText, and GloVe. Furthermore, we pre-trained a custom monolingual BERT model for Pashto and fine-tuned it for this task. We also explored the performance of multilingual BERT, XLM-R, by fine-tuning it. For model evaluation, we created a benchmark dataset, POLD, which is a collection of manually labeled tweets. Our experimental results show that the fine-tuned Pashto BERT model outperforms the other models, achieving an F1-score of 94.34% with an accuracy of

94.77%. Our investigation also revealed that the transformer models are comparatively more accurate than neural networks for Pashto offensive language detection. However, the performance difference is not very high, and LSTM with fastText embeddings can also achieve satisfactory results. We also noticed that the boundary between toxic and non-toxic textual content in Pashto is quite clear, which becomes more clear in a binary classification scenario. Therefore, for discrimination of Pashto text, particularly toxic *vs.* non-toxic, the classic neural networks such as LSTM and GRU can also be quite beneficial, both computationally and performance-wise.

This research is one of the pioneering works in Pashto NLP, and the contributions of this study are significant. Not only we proposed a model for offensive language detection but developed essential resources for NLP research in the low-resource Pashto language. All the resources and data are distributed publicly on GitHub and Kaggle. Additionally, the pre-trained models are included in the NLPashto (*Haq et al., 2023b*), which is an NLP toolkit for Pashto, available on the PyPi hub. All these resources can be used off-the-shelf, and we hope that they will facilitate and speed up future research in this domain.

### Funding
The authors received no funding for this work.

### Competing Interests
The authors declare there are no competing interests.

### Author Contributions
- Ijazul Haq conceived and designed the experiments, performed the experiments, analyzed the data, performed the computation work, prepared figures and/or tables, and approved the final draft.
- Weidong Qiu conceived and designed the experiments, analyzed the data, authored or reviewed drafts of the article, and approved the final draft.
- Jie Guo analyzed the data, authored or reviewed drafts of the article, and approved the final draft.
- Peng Tang performed the experiments, analyzed the data, performed the computation work, authored or reviewed drafts of the article, and approved the final draft.

### Data Availability
Data is available at Haq, Ijazul. (2023). Pashto Offensive Language Dataset [Data set]. Zenodo. https://zenodo.org/record/8195797.

### Supplemental Information
Supplemental information for this article can be found online at http://dx.doi.org/10.7717/peerj-cs.1617#supplemental-information.

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
