# Peer review of "Pashto offensive language detection: a benchmark dataset and monolingual Pashto BERT"

_PeerJ Computer Science, doi:10.7717/peerj-cs.1617_

## Round 0.1 · original submission · Major Revisions

We have now received the comments from the respected reviewers. Based on the comments, we are asking for a re-submission of the paper after the authors carefully address the comments.

In addition, I would suggest that the authors include a section on Limitations of the work, that should clearly summarize the limitations of the dataset and the classification model. As one reviewer identified that the tasks such as annotation are subjective and should be done by more than one person, it is important that the authors mention this in the manuscript as a limitation of the dataset, particularly, the subjective voting of the human annotators or the inherent bias if the annotation was done by a single person.

Reviewer 1 ·

Basic reporting

The authors used clear and unambiguous, professional English in the article. The literature provided was relevant and sufficient. The raw data was shared and the figures and tables were professional. The results were relevant and self contained.

Experimental design

The research questions were well-defined, relevant and meaningful.

Validity of the findings

All underlying data have been provided. Conclusions are well-stated and linked to original research questions.

Additional comments

1. There are some abbreviations which were not explained like. TF-IDF
2. In section 5.4 Error Analysis, The results are presented in Table 5, but Table 6 is referred instead. Please recheck it.

Reviewer 2 ·

Basic reporting

The research work mainly focused on the deployment of CNNs and RNNs to classify tweets either offensive or not, trained over the dataset developed by the authors themselves. Authors have to consider the following points prior to the acceptance of their work.
1. Very generic abstract: authors should completely revise the abstract part.
2. Authors should work on the write-up part, by removing all grammatical errors and vague statements. There are numerous statements in the manuscript where the claim of an author is not supported by any reference.
3. As Pashto has been spoken in many regions of southeast Asia, which dialect of it is targeted in your work? Furthermore, data collection is one of the most sensitive steps in any project, and there is no detail that how biases and sentimental vague statements are handled by the authors.
4. The random selection of tweets for training purposes might increase the chances of skewness to a certain class, which may tarnish all the claims/results made by authors. That is one of the major points of concern.
5. Author has been claiming that it is one of the first works, which targets offensive language detection in Pashto, which seems doubtful.
6. It would be better if authors incorporate some graphical illustration of a complete layout/structure of the work.
7. Authors have employed DL models, but no specification (or structural layout) of these models has been mentioned. A list of parameters, hyperparameters, and even the dataset splitting method is ambiguous.
8. There should be a statistical analysis of the obtained results and the significance of true positives/true negatives should be compared with the false positives/false negative results.

Experimental design

Insufficient information.

Validity of the findings

The validity of the findings must be statistically analyzed by the author, by mentioning the set of parameters/hyperparameters they used, so that the doubt of model deployment as a black box is vanished.

·

Basic reporting

Maybe, find the comments section.

Experimental design

Somehow, yes

Validity of the findings

yes

Additional comments

An interesting idea is presented in the article titled “Pashto Offensive Language Detection: A Benchmark Dataset and Monolingual Pashto BERT”. The authors have developed a benchmark dataset for experimental and evaluation purposes. In this paper, the authors have applied deep learning and tuned the BERT model for Pashto offensive language detection used on social media. The contents are presented in a comprehensive manner; however, I have a few concerns regarding the article:
1. During the development of the POLD dataset, what was the search query that was used to accumulate tweets from Twitter using Twitter API?
2. Provide a full description of the tweets (Twitter data) in the Dataset development section. How many of these were retweeted? how many were hashtags? Provide details of gender contribution and demographics locations (see help from)
(a) Khan, S., Khan, H. U., & Nazir, S. (2021). Offline Pashto characters dataset for OCR systems. Security and Communication Networks, 2021, 1-7.
3. In the Introduction section, the authors provided three objectives for their research work. However, a short description is needed along with each objective to improve readability and understandability. For example, in the second objective, the authors claimed that “we have developed…” So, how many records are there in the database, and how these records are distributed for the identification task?
4. In section 3.2.3 the authors claimed that the tweets are manually annotated. Do all the authors know the Pashto language? If not then how a single person annotated all the data? Mostly, in the annotation process, more than two people contribute. During a conflict, if more than half are agreed upon, a decision then it is followed. In this work, I didn’t find such a contribution. In short, this obligates your annotation task.
5. In the revised version can you generate a word-frequency diagram to show, what are the most “offensive” and “non-offensive” words or phrases used in the Tweets?
6. Also, can you generate a table to show what are the targeted people/user for offensive behavior? I mean can you perform this type of clustering using your model?

---

## Round 0.2 · Minor Revisions

Based on the comments received from the reviewers, I am glad to recommend the paper for acceptance should the authors be willing to incorporate the minor changes recommended by the reviewers. Overall, the paper is in better shape now and the authors have attempted to address the comments of the reviewers.

Reviewer 1 ·

Basic reporting

no comments

Experimental design

no comments

Validity of the findings

no comments

Additional comments

The revisions have been made and article is now up to the mark.

Reviewer 2 ·

Basic reporting

1. The manuscript has some persistent issues, which were pointed out during the first round of a review. A graphical or tabular view of employed techniques is needed, to replicate the work for further investigation in the future, as asked in comment # 7 in the previous version of the revision. Still, the setup is left ambiguous.

2. Authors have inferred information from confusion matrices, without making any statistical analysis to illustrate the significance of obtained results. A comprehensive statistical analysis is required before the acceptance of the final version of this paper.

Experimental design

NA

Validity of the findings

NA

·

Basic reporting

no comment

Experimental design

no comment

Validity of the findings

no comment

Additional comments

No

---

## Author Rebuttal · Round 0.2

Dear Editor

Thank you for giving us the opportunity to submit a revised draft of our manuscript titled **"Pashto offensive language detection: a benchmark dataset and monolingual Pashto BERT"** to the Peer J Computer Science journal. We appreciate the time and effort you have dedicated to reviewing our work. Your feedback has been invaluable in improving the quality of our manuscript. We have carefully considered the comments and have made substantial revisions. In the revised manuscript, we have incorporated the suggested changes, and here we present a point-by-point response to your comments and concerns:

## Editor comments (Hazrat Ali)

We have now received the comments from the respected reviewers. Based on the comments, we are asking for a re-submission of the paper after the authors carefully address the comments.

In addition, I would suggest that the authors include a section on Limitations of the work, that should clearly summarize the limitations of the dataset and the classification model. As one reviewer identified that the tasks such as annotation are subjective and should be done by more than one person, it is important that the authors mention this in the manuscript as a limitation of the dataset, particularly, the subjective voting of the human annotators or the inherent bias if the annotation was done by a single person.

**Response:** Dear editor, we agree with your opinions regarding the manual annotation of the dataset. We acknowledge the subjective nature of tasks like annotation and the potential for bias in the labeling process. Therefore, we have involved a total of 5 experts in the manual annotation process, and the annotation procedure is detailed in Section 3.2.3 of the revised manuscript. We also explained in the first paragraph of that section why we chose a limited number of participants rather than crowdsourcing, and also mentioned the guidelines followed to minimize bias in the dataset annotation.

Dear editor, the dataset annotation was the most challenging and laborious part of this research. It was also the most time-consuming task, hence we hired paid professionals (not volunteers) to accomplish it. Our lab has spent more than 5K RMB (≈200K PKR) to remunerate the experts who participated in the manual annotation to speed-up the annotation process with the best quality.

As for the limitations of the model, they are few, but relevant and deserve consideration. Hence, we have included them in Section 5.4 of the revised manuscript, where we discuss common errors and mistakes of the model.

## Reviewer 1 (Anonymous)

1    There are some abbreviations which were not explained like. TF-IDF

**Response:** Thank you for bringing this to our attention. In the revised manuscript, we have ensured that all abbreviations, including TF-IDF, are clearly defined upon their first mention.

2    In section 5.4 Error Analysis, The results are presented in Table 5, but Table 6 is referred instead. Please recheck it.

**Response:** We have proofread the whole manuscript again, and addressed the referencing issues.

## Reviewer 2 (Anonymous)

1   Very generic abstract: authors should completely revise the abstract part.

**Response:** In response to this comment, we have thoroughly revised the abstract to provide a more specific and concise summary of our research.

2   Authors should work on the write-up part, by removing all grammatical errors and vague statements. There are numerous statements in the manuscript where the claim of an author is not supported by any reference.

**Response:** We apologize for any spelling or grammatical mistakes and vague statements that may have been present in the initial submission. In the revised manuscript, we have meticulously addressed all grammatical errors and vague statements to enhance the clarity and coherence of the content. Furthermore, we have reinforced each claim with appropriate references.

3   As Pashto has been spoken in many regions of southeast Asia, which dialect of it is targeted in your work? Furthermore, data collection is one of the most sensitive steps in any project, and there is no detail that how biases and sentimental vague statements are handled by the authors.

In this comment, the reviewer has asked two important questions:

i.   Which Pashto dialect is targeted in you work?
     **Response:** As mentioned by the reviewer, Pashto has several dialects, but we haven't mentioned any specific dialect in our paper. The reason is that, the dataset is a collection of tweets, which encompass all the dialects, because the seed list we used to query Twitter include words and phrases from all the dialects. And shown in the figure bellow, we used to set the language to Pashto (lang='ps') in our quarry, where in response the Twitter API used to return all Pashto tweets where the keyword was found, regardless of the dialects. Therefore, the models we trained on this dataset are also universal and perform equally well in all the dialects. We have clarified this ambiguity in section 3.2.1 of the revised manuscript.

```python
df = pd.read_csv(f'dataset/keywords.csv', header=None)
seed_list = df.iloc[:,0]

collected_tweets = []
for index, keyword in enumerate(seed_list):
    try:
        tweets = api.search_tweets(q=keyword, count=200, lang='ps')
        for t in tweets:
            collected_tweets.append(t._json)
        print(index, q, len(tweets))
    except:
        print(f'\nFailed: {index}\t{keyword}\n')
```

ii.  How biases and sentimental vague statements are handled by the authors?
     **Response:** We have addressed these concerns; please refer to our response on editor's comments (above in this document).

| 4 | The random selection of tweets for training purposes might increase the chances of skewness to a certain class, which may tarnish all the claims/results made by authors. That is one of the major points of concern. |
|---|---|

**Response:** We acknowledge the possibility of skewness in class distribution due to random selection. To mitigate this issue, we have carefully stratified the data to ensure a proportional representation of both offensive and non-offensive tweets in the training dataset. In the last two sentences of Section 3.2.4 of the revised manuscript, we have provided a more explicit explanation of data splitting and balancing techniques.

| 5 | Author has been claiming that it is one of the first works, which targets offensive language detection in Pashto, which seems doubtful. |
|---|---|

**Response:** We agree with the reviewer's comments. In the revised manuscript, all the doubtful statements (including this one) are re-written with more clarity.

| 6 | It would be better if authors incorporate some graphical illustration of a complete layout/structure of the work. |
|---|---|

**Response:** Thanks for the suggestions. In the revised manuscript, we have incorporated the graphical illustration of our work in Figure 1.

| 7 | Authors have employed DL models, but no specification (or structural layout) of these models has been mentioned. A list of parameters, hyperparameters, and even the dataset splitting method is ambiguous. |
|---|---|

**Response:** Thanks for the suggestions. In the revised manuscript we have included a graphical illustration, which shows the high-level structural layout of the DL models in Figure 4. The components which are common in all the DL models are discussed in the 1st paragraph of Section 4.1.3, and the individual specifications of each model are discussed in the 2nd paragraph of the same Section. The architecture and hyper-parameters setup of all the DL models is given in Table 3. And a summary of the dataset and splitting method is presented in Section 3.2.4 and Table 1, where the same splitting strategy is kept uniform for all the DL and transfer learning models.

| 8 | There should be a statistical analysis of the obtained results and the significance of true positives/true negatives should be compared with the false positives/false negative results. |
|---|---|

**Response:** Thanks for the suggestions. We have edited the first paragraph of Section 5.4, as well as updated Figure 10 in the revised manuscript to accumulate such statistics.

## Reviewer 3 (Muhammad Rafiul Biswas)

| 1 | During the development of the POLD dataset, what was the search query that was used to accumulate tweets from Twitter using Twitter API? |
|---|---|

**Response:** We appreciate the reviewer's interest in technical details. This question is answered in response to the 3rd comment of reviewer 2, as well as elaborated in Section 3.2.1 of the revised manuscript.

2    Provide a full description of the tweets (Twitter data) in the Dataset development section. How many of these were retweeted? how many were hashtags? Provide details of gender contribution and demographics locations (see help from) (a) Khan, S., Khan, H. U., & Nazir, S. (2021). Offline Pashto characters dataset for OCR systems. Security and Communication Networks, 2021, 1-7.

**Response:** We appreciate the reviewer's interest in additional data details. However, in this research, our main focus is on developing an AI model for text classification and evaluating its performance. Providing details on tweets types and demographics is beyond the scope of this study. We would like to mention that the dependence on Twitter data for our research was not strictly necessary, and we could have collected data from any social media platform. However, we encountered limitations in accessing alternative sources to gather relevant textual content, so we opted to gather data from Twitter. Nevertheless, we appreciate the interest in data details and will consider them for future investigations.

3    In the Introduction section, the authors provided three objectives for their research work. However, a short description is needed along with each objective to improve readability and understandability. For example, in the second objective, the authors claimed that "we have developed..." So, how many records are there in the database, and how these records are distributed for the identification task?

**Response:** Thank you for the suggestion. A short description of each objective is given in the introduction section, while the contributions are summarized in bullets form at the end of the introduction section.

4    In section 3.2.3 the authors claimed that the tweets are manually annotated. Do all the authors know the Pashto language? If not then how a single person annotated all the data? Mostly, in the annotation process, more than two people contribute. During a conflict, if more than half are agreed upon, a decision then it is followed. In this work, I didn't find such a contribution. In short, this obligates your annotation task.

**Response:** We have addressed these concerns. Please refer to our response on editor's comments (above in this document).

5    In the revised version can you generate a word-frequency diagram to show, what are the most "offensive" and "non-offensive" words or phrases used in the Tweets?

**Response:** Certainly, in the revised manuscript, we have generated a word-frequency diagram represented in Figure 3 to illustrate the most commonly used "offensive" and "non-offensive" words or phrases in Pashto Tweets.

6    Also, can you generate a table to show what are the targeted people/user for offensive behavior? I mean can you perform this type of clustering using your model?

**Response:** Certainly, the model can be used to identify the victims of offensive behavior, which is only one of the countless applications/use-cases of the proposed model. The model demonstrates potential applications in various domains, including sociology and social sciences, where it can be used to identify people/users who are the victims of cyberbullying, harassment or hate speech etc. It can also be used to analyze users' sentiments towards social media posts or online products and services. However, the current research is primarily focused on developing the model. While we appreciate the reviewer's suggestions, exploring these diverse applications falls outside the scope of this study, and detailed discussion on the applications of the model may require significant additional research, analysis, and data collection, leading to a substantial increase in the paper's length.

---

## Round 0.3 · accepted · Accept

The authors have improved the manuscript. I am glad to recommend the paper for acceptance.